# Impact of Selective Dry Cow Therapy on Antimicrobial Consumption, Udder Health, Milk Yield, and Culling Hazard in Commercial Dairy Herds

**DOI:** 10.3390/antibiotics12050901

**Published:** 2023-05-12

**Authors:** Zyncke Lipkens, Sofie Piepers, Sarne De Vliegher

**Affiliations:** 1Milk Control Center Flanders, 2500 Lier, Belgium; zyncke.lipkens@mcc-vlaanderen.be; 2M-team & Mastitis and Milk Quality Research Unit, Department of Internal Medicine, Reproduction and Population Medicine, Faculty of Veterinary Medicine, Ghent University, 9820 Merelbeke, Belgium; sofie.piepers@ugent.be

**Keywords:** selective dry cow therapy, test-day SCC, cow performance, antimicrobial consumption

## Abstract

The main objective of the study was to evaluate whether or not implementing selective dry cow therapy (SDCT) on commercial dairy farms reduces antimicrobial consumption without negatively affecting future performances when compared to blanket dry cow therapy (BDCT). Twelve commercial herds in the Flemish region of Belgium with overall good udder health management were enrolled in a randomized control trial, including 466 cows that were assigned to a BDCT (*n* = 244) or SDCT (*n* = 222) group within herds. Cows in the SDCT group were dried off with internal teat sealants combined or not with long-acting antimicrobials according to a predefined algorithm based on test-day somatic cell count (SCC) data. Total antimicrobial use for udder health between drying off and 100 days in milk was significantly lower in the SDCT group (i.e., a mean of 1.06 defined the course dose) compared to the BDCT group (i.e., a mean of 1.25 defined the course dose), although with substantial variation between herds. Test-day SCC values, milk yield, and the clinical mastitis and culling hazard in the first 100 days in milk did not differ between the BDCT and SDCT groups. SCC-based and algorithm-guided SDCT is suggested to decrease the overall use of antimicrobials without jeopardizing cows’ udder health and milk yield.

## 1. Introduction

For more than 70 years, antimicrobials have been used as an important tool to control bacterial infection in animal husbandry. Unfortunately, studies have documented a positive correlation between antimicrobial consumption and the emergence of antimicrobial resistance in animal-associated bacteria [1,2,3]. Therefore, efforts are demanded to enhance the prudent use of antimicrobials in veterinary medicine. The intramammary application of long-acting antibiotics to all cows at dry-off (i.e., blanket dry cow therapy (BDCT)) is one of the main contributors to antimicrobial consumption in dairy cows [4]. Dry-off is a transitional phase between lactation during which the cow produces milk and the actual dry period. For dairy cows, the dry period represents a time of rest in which the cow is not producing milk, and is essential to the success of upcoming lactation and further reproduction performances. In the dairy sector, selective dry cow therapy (SDCT), i.e., only administering antimicrobials to cows when they are likely infected at drying off, is one of the recommended strategies to meet the justified public demands. With the new European Regulation 2019/6 on Veterinary Medicines that came into force on 28 January 2022, the prophylactic use of antibiotics is nowadays no longer authorized; thus, BDCT is no longer authorized either.

Drying off major pathogen-infected cows without antimicrobials has a potential negative impact on future cow welfare and performances; e.g., in [5,6,7]. Therefore, it is preferable to differentiate major pathogen-infected cows from uninfected and minor pathogen-infected cows at drying off. To detect intramammary infections (IMI), somatic cell count (SCC) is used as an easy-to-use and cheap yet reliable alternative for conventional bacteriological culturing, which is labor-intensive and time-consuming. On-farm and on-practice alternatives are promising but not as widespread and studied compared to (combinations of) thresholds based on the SCC for which high negative predictive values have been demonstrated. In 2019, we reported that combining test-day SCC information from the three last milk recordings before drying off allowed the accurate distinction of uninfected/minor pathogen-infected cows from major pathogen-infected cows at drying off, resulting in slightly different predictive values than provided by a single last-test-day SCC [8]. Additionally, the test characteristics and predictive values were modified by the herd’s bulk milk somatic cell count (BMSCC) and the cow’s milk yield (MY) and parity [8]. Combining this easily accessible dairy herd improvement (DHI) milk recording information in an algorithm to selectively allocate antimicrobial treatment at the cow level would be helpful, though it has not yet been applied in the field.

Despite the fact that BDCT has been banned in the EU since the beginning of 2022, SDCT is not yet widely adopted in the EU (with variation within and between regions and countries) as there is still the perception that SDCT can jeopardize the future performance of cows and might increase antimicrobial use during lactation. In this regard, different studies have reported the absence of important differences between treatment groups (BDCT vs. SDCT) under the specific conditions of the trials [9], when assessing test-day MY [10,11,12,13], clinical mastitis (CM) hazard [7,11,12,13,14] and culling hazard [12,13] during subsequent lactation. In previous studies, SDCT has been applied using SCC information derived from the last DHI recording in combination with parity and the clinical appearance of the cow and teats [7], SCC data derived from the three last DHI recordings in combination with CM history in previous lactation [12] or monthly DHI recordings covering the complete lactation period [15,16]. Furthermore, in some studies, a BMSCC of <250,000 cells/mL [11,14,15] and/or administration of an internal [11,14,15,17] or external teat sealant [12] to all quarters of all cows in the herd was required. Another major consideration is that in some studies, the sampling and administration of dry cow tubes was conducted by well-trained study technicians and not by the dairy producer (on smaller herds) or farm staff. Differences in herd and cow inclusion criteria as well as in study implementation make it obviously difficult to straightforwardly compare results between studies and to extrapolate the different findings to other herds, especially if they are located in other regions and countries. Still, the more studies conducted in different regions and countries, the more precise the insights into what dairy producers can expect when shifting from BDCT to SDCT.

The main objective of this study was to estimate the expected impact of SDCT on total antimicrobial consumption for udder health between drying off and the first 100 days in milk (DIM) on commercial dairy farms in the Flemish region of Belgium with a geometric mean BMSCC of ≤250,000 cells/mL for a period of six months before enrollment in the study. Additionally, the potential impact of SDCT compared to BDCT on test-day SCC and MY values in the first 100 DIM, and the CM and culling hazard in the subsequent lactation period was studied.

## 2. Results

### 2.1. Herd Descriptive Results

On average, the herds housed 90 cows (ranging from 56 to 139 cows) during the study period and had an average dry period length (i.e., number of days between drying off and calving) of 46 days (ranging from 40 to 59 days). The test-day MY at the last DHI record before drying off was normally distributed with a mean of 24.3 kg per day (standard deviation of 7.6 kg). The geometric mean BMSCC in March 2017 (i.e., the start of the study), calculated based on at least four records per month, was 145,000 cells/mL (ranging from 84,000 to 195,000 cells/mL) while the mean 305-day milk production was 9168 kg (ranging from 5514 to 12,213 kg). More detailed information per herd and per treatment group can be found in Table 1 and Table 2, respectively.

Cow characteristics prior to drying off per treatment group can be found in Table 1. The three last-test-day SCC values prior to drying off were obtained up until 151 days before drying off with an interval between DHI records ranging from 23 to 59 days (averaging at 37.7 days).

### 2.2. Bulk Milk Somatic Cell Count Strata

As the geometric mean BMSCC over the last six months of each herd varied during the study period, herds were able to shift from a so-called low- to a high-BMSCC herd or vice versa. At the start of the study, in March 2017, four herds (herds 2, 3, 10 and 12; Table 2) were defined as high-BMSCC herds, and remained high-BMSCC herds until October 2017, when the last cow was dried off in the study. The same observation was made for three low-BMSCC herds (herds 1, 4 and 11; Table 2) that remained low-BMSCC herds until October 2017. Yet, the remaining five herds (herds 5, 6, 7, 8 and 9; Table 2) were initially low-BMSCC herds and became high-BMSCC herds for at least one month. Importantly, during the entire field study, the geometric mean BMSCC threshold of 250,000 cells/mL was never exceeded by any of the herds and therefore all herds remained in the study.

### 2.3. Differentiation of Major Pathogen-Infected Cows from Minor Pathogen-Infected or Uninfected Cows at Drying off in the SDCT Group

The infection status at drying off was determined by the algorithm [7] (Figure 1) for the 222 cows in the SDCT groups. Eighteen cows were considered major pathogen-infected by default because of the occurrence of CM between the last-test-day SCC and drying off (*n* = 7) or the unavailability of one or more of the three last-test-day SCC values before drying off (*n* = 11), and therefore received long-acting intramammary antimicrobials at drying off (Figure 2). Of the remaining 204 cows in the SDCT group, 108 (52.9%) cows were part of a high-BMSCC herd at drying off, of which 18 out of 45 (40.0%) primiparous cows and 5 out of 63 (7.9%) multiparous cows were considered to be minor pathogen- infected or uninfected at drying off according to the algorithm. Ninety-six (47.1%) cows were part of a low-BMSCC herd at drying off, of which 28 out of 32 (87.5%) primiparous cows and 24 out of 64 (62.5%) multiparous cows were considered to be minor pathogen-infected or uninfected at drying off (Figure 2). All in all, 75 out of the 222 SDCT cows (33.8%) only received internal teat sealants at drying off, with a between-herd variation of 6.2 to 73.9% (Figure 3).

### 2.4. Monitoring Period

Cows were monitored between the time of drying off until 100 DIM (or culling), with a mean duration of 142 days in both treatment groups (ranging from 14 to 209 days in the BDCT group with a standard deviation of 23 day and ranging from 1 to 221 days with a standard deviation of 27 days in the SDCT group). The mean dry period length was 45 days (ranging from 5 to 109 days with a standard deviation of 16 days) and 48 days (ranging from 2 to 121 days with a standard deviation of 17 days) in the BDCT and SDCT group, respectively (non-parametric Mann–Whitney U test; *p* = 0.08).

### 2.5. Antimicrobial Consumption

From drying off until 100 DIM, cows in the SDCT group received significantly less antimicrobials (in defined course doses (DCD)) in relation to udder health than cows in the BDCT group did (a reduction of an average of 22% in the SDCT group compared to the BDCT group when comparing the total antimicrobial consumption for udder health for both groups; *p* < 0.001) (Table 3; Figure 4). Dry cow therapy accounted for most of the antimicrobials used in both the SDCT group with 149 nDCD_dry cow_ and in the BDCT group with 245 nDCD_dry cow_, representing 63.0 and 80.9% of the total nDCD administered for udder health in each treatment group, respectively (Table 3).

### 2.6. Performance after Calving

#### 2.6.1. Test-Day SCC and Milk Yield

After calving, 1150 milking records of 450 cows were available, ranging between 1 to 4 DHI records per cow with an interval between DHI records ranging from 25 to 59 days, averaging at 36 days. Test-day LnSCC and MY values were significantly associated with DIM (*p* < 0.001), quadratic DIM (*p* < 0.001), and parity (*p* < 0.001), but not with the treatment group. Test-day SCCs and MY (least-square means of 3.54 and 38.8 kg, respectively) during the subsequent lactation of cows that received SDCT were not significantly different from those of cows that received BDCT (least-square means of 3.53 and 38.4 kg, respectively) (*p* = 0.89 and *p* = 0.54, respectively) (Table 4 and Table 5, respectively). Test-day milk yield decreased with an increasing LnSCC (*p* < 0.001) (Table 5).

#### 2.6.2. Clinical Mastitis Hazard

The CM hazard did not differ (*p* = 0.48) between cows that received SDCT and cows that received BDCT (Table 6 and Figure 5). In total, 82 cows developed CM between the time of drying off and 100 DIM, of which 42 cows (18.9%) were part of the SDCT group (*n* = 222) and 40 cows (16.4%) were part of the BDCT group (*n* = 244). Primiparous cows had a lower likelihood of developing CM in the first 100 DIM than multiparous cows were (*p* = 0.004).

In the SDCT group, 42 cows developed CM between drying off and 100 DIM. One cow developed CM caused by *Staphylococcus aureus* 35 days before calving (a dry period length of 53 days), while another cow developed CM 2 days before calving (unknown cause due to non-sampling by the dairy producer). For the other 40 cows, CM was observed within the first 100 DIM at a median DIM of 21. Eleven out of the 42 cows (26.2%) with CM did not receive antimicrobials at drying off, including the cow with *S. aureus* CM 35 days before calving. Additionally, when comparing the group of cows that received antimicrobials at drying off (147 out of 222 SDCT cows) with the group of cows that did not receive antimicrobials at drying off (75 out of SDCT 222 cows), 31 out of 147 cows (21.1%) that received antimicrobials and 11 out of 75 (14.7%) that did not receive antimicrobials at drying off developed CM between drying off and the first 100 DIM (ratio of 1.4). In the BDCT group, 40 cows were affected by CM at a median DIM of 38.5 days, and obviously all of them received antimicrobials at drying off (Table 7).

#### 2.6.3. Culling Hazard

The hazard of culling did not differ between cows that received SDCT and cows in the BDCT group (*p* = 0.24) (Table 6 and Figure 6). In total, 32 cows were culled from time of drying off until 100 DIM in the subsequent lactation, of which 19 cows (8.6%) were in the SDCT group (*n* = 222) and 13 cows (5.3%) were in the BDCT group (*n* = 244). Primiparous cows had a lower likelihood of being culled than multiparous cows did (*p* = 0.01).

In the SDCT group, three cows in total were culled during the dry period due to lameness (*n* = 1), non-pregnancy (*n* = 1) and abortion (*n* = 1). The other 16 cows were culled at a median DIM of 53 days, of which six (37.5%) were culled due to udder health problems at a median DIM of 62.5 days. All cows had received antimicrobials at drying off, with the exception of one cow that was culled because of a non-udder-health-related reason.

In the BDCT (*n* = 244), one cow was culled after she aborted. Twelve cows were culled after calving at a median DIM of 41.5 days, of which four cows (33.3%) were culled due to udder health problems at a median DIM of 79.5 days.

## 3. Discussion

The aim of this randomized clinical trial was to evaluate the implementation of SDCT in commercial dairy herds using a predefined algorithm based on the BMSCC, milk recording data and CM history before drying off to select cows to be treated with long-acting antimicrobials or not. Cows were randomly divided into two groups based on ear tag number within herds, allowing for a good estimation of the impact of shifting from BDCT to SDCT under the same herd management.

The algorithm that was used in this study aimed at differentiating between minor pathogen-infected and uninfected cows (i.e., those not in need of antimicrobial treatment) from major pathogen-infected cows (i.e., those in need of long-acting antimicrobials), with great certainty. For those reasons, (1) only herds with good udder health were enrolled, defined as a geometric BMSCC during 6 months before drying off of ≤250,000 cells/mL; (2) all cows received internal teat sealants at drying off; (3) cows affected by CM between the last DHI record and the time of drying off were considered major pathogen-infected by default and in need of antimicrobial treatment; (4) the three last-day SCC values before drying off, obtained via DHI recording, were included and the SCC thresholds were chosen as a function of the highest negative predictive values [8]. In this way, a significant reduction in overall antimicrobial use for udder health of 22% on average was obtained when cows were selectively allocated for antimicrobial treatment, without jeopardizing the test-day SCC and MY values, CM risk, and culling hazard.

The negative predictive value of the applied algorithm was estimated to be 85.2%, which is in accordance with the findings of our previously published study (89.9%) [8]. A total of 9 out of 61 cows considered to be uninfected/minor pathogen-infected based on the culture-independent algorithm and of which the true IMI status at drying off could be determined were actually major pathogen-infected at drying off (i.e., with *Staphylococcus aureus* (*n* = 1), Gram-negative bacteria (*n* = 1), Gram-negative bacteria and *Streptococcus dysgalactiae* (*n* = 1), *Streptococcus uberis* (*n* = 3), and other esculin-positive streptococci than *Streptococcus uberis* (*n* = 3)). Still, the geometric mean SCC of those 9 cows at the third-last, second-last and last DHI record before drying off ranged between 32,000 and 39,000 cells/mL, while the SCC of none of these cows ever exceeded 125,000 cells/mL at any DHI record. This implies that either an IMI causing subclinical mastitis occurred between the last DHI record and drying off or the cow showed no inflammatory response to the presence of the bacteria. Despite the fact that these cows did not receive antimicrobials at drying off, only 2 out of the 9 cows developed CM during early lactation. One cow was infected with *Escherichia coli* at 59 DIM, which likely persisted during the dry period as *E. coli* was also found at drying off, although this was not confirmed using strain typing. Of course, it cannot be excluded that the cow contracted a new *E. coli* IMI after calving, which resulted in CM at 59 DIM. Post-calving culture results were unfortunately not available though would have allowed tracing back the origin of this and other cases of CM after the dry period and for unravelling whether or not the cows were erroneously considered infected with a major pathogen at drying off. The other cow, on the contrary, was infected with *S. aureus* at drying off and contracted a new yeast infection at 69 DIM.

The approach of only treating likely major pathogen-infected cows, and thus aiming at leaving uninfected but also minor pathogen-infected cows untreated, is rather uncommon, yet comes with a substantial additional reduction in antimicrobial use. Most studies administered long-acting antimicrobials to all cows that were infected at dry-off, independently of the pathogen that was involved; e.g., [11,12,18]. Especially in herds with low BMSCC values, non-*aureus* staphylococci IMI may be an important contributor to the total number of somatic cells in the bulk milk [19]. The latter observation favors the approach applied in these studies. Still, the NAS species that have been most frequently identified as the cause of bovine IMI also induced the highest increase in [20,21,22], which in one study was even comparable to the increase caused by *S. aureus* [20]. Given all of this, it is not unlikely that cows infected with *S. chromogenes*, *S. simulans*, *S. epidermidis*, *S. xylosus* or *S. haemolyticus* in at least one quarter will be falsely considered major pathogen-infected by the proposed algorithm and thus be treated with long-acting antimicrobials. Interestingly, in Norway, a country that is known for its prudent antimicrobial use [23], only treating major pathogen-infected cows at drying off has been recommended for many years and comes with overall good udder health results [24]. Our study follows the Norwegian guidelines yet protects udder health as much as possible via an algorithm striving for the highest negative predictive values to identify uninfected and minor pathogen-infected cows with great certainty, and via aiming at treating the bacteria known to harm udder health (i.e., major pathogens). Moreover, only easily-accessible data that can be made available on a routinely basis on every dairy farm are used (i.e., DHI and BMSCC data), making the algorithm presented easy to be used and fine-tuned, for instance to increase positive predictive values and thus establishing the certainty of only treating infected cows with antimicrobials.

The estimates of the effect of SDCT on both test-day SCC and MY values rather lacked some precision because of the high variation among cows in test-day SCC values and test-day MY values in both groups relative to the sample size. Still, the difference in test-day SCC between SDCT and BDCT appeared to be negligible and on average even slightly in favor of the SDCT group. The negligible effects on the SCC during early lactation are in accordance with the findings reported in other recent trails using external [10] or internal teat sealants [12,13]. The observed lower SCC in the SDCT group compared to the BDCT group might be partially linked to the numerically higher incidence of CM during the ensuing lactation period. The odds of being culled were 2.3 times higher for cows that experienced CM compared to cows that did not experience CM. We hypothesize that these cows with a potentially higher test-day SCC values were culled and thus no longer contributed to the dataset, explaining at least to some extent the somewhat lower SCC on average in the SDCT group. On the other hand, the test-day SCC of cows in the SDCT group that did not receive antimicrobials at drying off was numerically lower compared to that of the cows in the same group that received antimicrobials, although they appeared to be less susceptible for developing CM in the first 100 DIM.

Due to the rather imprecise estimates for the effect of SDCT on test-day MY, a lower test-day MY during the next lactation period in the case of SDCT cannot be ruled out based on our data. Still, the cows in the SDCT group showed higher average test-day MY values in the first 100 DIM compared to the cows in the BDCT group, which is in accordance with the observations of McParland et al. [25]. In the latter study, low-SCC cows dried off with an internal teat sealant alone produced 0.67 kg more milk per day on average throughout the entire lactation period compared to the cows that were dried off with a combination of long-acting antimicrobials and an internal teat sealant. The slightly higher test-day MY in the SDCT group compared to that in the BDCT group found in our study is difficult to explain. The pre-diagnosis milk drop in case of CM, which was previously observed [26,27], might have been more detrimental for test-day MY in the BDCT group than in the SDCT group as CM occurred about 20 days earlier in the SDCT group than it did in the BDCT group. Of course, coincidence cannot be ruled out either as the estimated differences were small and rather imprecise.

However, the risk of CM and the culling hazard after drying off were numerically not significantly higher for cows in the SDCT group than they were for cows in the BDCT group. The findings are in agreement with the results reported by Cameron et al. [11] and Vasquez et al. [12]. Scherpenzeel and co-workers (2014), who enrolled 97 commercial dairy herds in a large field study, reported significant differences between treated and untreated quarters within cows, which was at least partially driven by the larger sample size. Of course, the significant differences might also be due to the fact that no internal teat sealant (known to be a highly efficient dry cow management tool) was applied in that study, other than the different selection criteria to leave cows untreated, making comparison difficult. In the SDCT group, about 75% of the cows developing CM received antimicrobials at drying off as they were considered infected with a major pathogen at that time. Furthermore, as dairy producers were not blinded to the dry cow treatment and only just introduced SDCT in a part of their herd, it is imaginable that cows in the SDCT group were monitored more closely, which could explain the numerically higher amount of CM in the SDCT group compared to the BDCT group.

Of course, SDCT will only become successful if, in addition to maintaining good udder health and MY, a substantial reduction in the total antimicrobial use for udder health is accomplished as this is the main argument for shifting from blanket to selective dry cow therapy. This implies that the number of antimicrobials used for treating additional cases of mastitis in the next lactation period, possibly as a consequence of not using antimicrobials during the dry period, does not outweigh the reduction at drying off (also reported by Scherpenzeel et al. [7] and Vasquez et al. [12]). Overall, a reduction of 22% on average of the total antimicrobial use for udder health was achieved. An economic analysis was beyond the scope of this study, although the results could have been helpful to reassure dairy producers and convince them to make the switch from BDCT to SDCT [28,29] A recent study published by Scherpenzeel et al. [30] stipulated that the economic cost of mastitis is affected more by the BMSCC and the incidence of CM than by the dry-off strategy. Additionally, for all evaluated types of herds (low vs. high BMSCC and low vs. high incidence of CM), SDCT was economically more beneficial than BDCT was. Still, economic profits from SDCT were larger if the BMSCC and CM incidence were lower.

Implementing our algorithm in a larger scale-field study including a substantially higher number of commercial dairy herds would enable one to identify smaller differences in performances between cows in the SDCT group and cows in the BDCT group as statistically significant since a larger sample size including more cows within more different herds would obviously further increase the power of the study. A larger sample size would also allow the better external validity of the results as more different herd management types (e.g., milking parlor, bedding material, herd size, antimicrobials administered, etc.) would be included. After all, the dairy herds that met the inclusion criteria and that were volunteered to enroll in this SDCT trial can obviously not be considered average herds. All herds enrolled in the study participated in the DHI scheme whereas in the Flemish region of Belgium only 50% of the dairy herds do so. Additionally, the average BMSCC in the Flemish region of Belgium was around 220,000 cells/mL at the time that the study was conducted. Based on the median value of 157,000 cells/mL for the BMSCC, we can conclude that the herds enrolled in our study most likely performed better when it came to mastitis management than the average dairy herd in the Flemish region of Belgium performs.

This study combines a number of requirements that were found to be successful in other studies (such as a herd’s BMSCC, individual SCC and CM history), while keeping the implementation simple in any commercial dairy herd.

## 4. Materials and Methods

### 4.1. Herds and Animals

A convenience sample of twelve commercial dairy farms milking a total of 1067 dairy cows were recruited on a voluntary basis to participate in a randomized control trial conducted from March 2017 to April 2018. For practical reasons, a minimum herd size of 50 lactating cows was required as well as was a herd situated within a 50 km radius from the Faculty of Veterinary Medicine of Ghent University (Merelbeke). All herds had to participate in the local DHI program of the cooperative cattle improvement organization (CRV, Sint-Denijs-Westrem, Belgium) on a four- to six-weekly basis, and were required to have a geometric mean BMSCC of ≤250,000 cells/mL [31] for a period of six months before enrollment in the study. Before the start of the study, all herds had BDCT applied. All herds were milked twice daily using a conventional milking parlor and cows were calved year-round.

### 4.2. Study Design

All 1067 cows in the 12 herds were eligible and were allocated to a BDCT or SDCT group within herds at the start of the study period based on their ear tag number (Table 1) in accordance to Borchardt et al. [32]. Cows with an odd ear tag number received BDCT at drying off (BDCT group; 244 cows or 52.4%)]; they were administered long-acting antimicrobials (the same product as that used in the herd before the start of the study) and internal teat sealants at drying off in all quarters. Cows with an even ear tag number were selectively dried off with or without long-acting antimicrobials (SDCT group; 222 cows or 47.6%) depending on their estimated infection status using a predefined algorithm (based on Lipkens et al. [8]), yet always received internal teat sealants in all quarters at drying off.

The dry cow antimicrobials used were decided by the producers with the supervision of their veterinarian, as this was carried out in daily practice. Additionally, the antimicrobial compound used at drying off could be modified from one cow to another within a herd, as decided by the dairy producer. Most herds (9 out of 12) made use of broad-spectrum long-acting antimicrobial dry cow tubes, while the remaining three herds used narrow-spectrum long-acting antimicrobial tubes (Table 1). As dry cow therapy was administered by the producers, their drying off technique including udder preparation, the insertion of the tubes, hand hygiene, etc. was evaluated and corrected when needed before the start of the trial to ensure aseptic infusion.

In addition to long-acting dry cow tubes, producers were allowed to administer antimicrobials systemically at drying off, but only for cows in the BDCT group or only if cows were considered to be infected with a major pathogen in the SDCT group based on the culture-independent algorithm. In three herds, a total of seven cows received additional systemic antimicrobials at drying off, four of them being part of the herds’ BDCT group and three of them being part of the herds’ SDCT group.

A treatment effect was estimated by comparing total antimicrobial consumption for udder health and the CM hazard starting at drying off until 100 DIM, test-day SCC and MY values, and culling hazard in the next lactation period until 100 DIM.

### 4.3. Differentiation of Suspectedly Infected Cows from Uninfected Cows at Drying off in the SDCT Group

Cows with CM between the last test-day and the day of drying off were considered in need of long-acting intramammary antimicrobials, as were cows for which one or more of the three last test-day SCC records were lacking.

For all other cows, the infection status was estimated around drying off using an algorithm based on the three last-test-day SCC values prior to drying off. Additionally, the estimated herd’s BMSCC (the so-called high-BMSCC herd (a calculated geometric mean BMSCC over 6 months of ≥157,000 cells/mL) vs. the so-called low-BMSCC herd (a calculated geometric mean BMSCC over 6 months of <157,000 cells/mL)) at the time of drying off as well as the parity of the cow (primiparous vs. multiparous cows) were taken into account. The threshold of 157,000 cells/mL was chosen based on the results obtained in a previous study including 15 dairy herds that were similar to the ones included in this study. The median BMSCC of the herds involved in that study was 157,000 cells/mL [8]. Additionally, from every dairy cow, milk samples for bacteriological culturing were collected at dry-off as described in Lipkens et al. [8]. The data collected in the current study allowed us to validate the test characteristics and predictive values obtained in our previous study [8] by recalculating these numbers of the algorithm vs. the bacteriological culture results obtained at drying off.

### 4.4. Data Collection and Treatment Outcomes

#### 4.4.1. Antimicrobial Consumption

Detailed antimicrobial use records were kept by the producers when antimicrobials were used for udder health issues (i.e., dry cow therapy or (clinical) mastitis treatment) including cow identification, product name and volume, day, route of administration and indication.

During the monitoring period, and thus from time of drying off until 100 DIM (or until culling if the cow was culled before 100 DIM), the total number of DCD (nDCD) was calculated. The nDCD was calculated for each cow (4), in accordance with the European Medicines Agency recommendations (EMA) [33,34] based on the usage per cow. The usage per cow was calculated based on the product-level-defined daily dose, as provided by AMCRA, the Belgian center of expertise on antimicrobial consumption and resistance in animals. For injectables, calculations were adjusted for a standard body mass of 425 kg (1). Four dry cow intramammary injectors were classed as 1 DCD (nDCD_dry cow_), regardless of antimicrobial active ingredient, concentration or duration of action (2). For all intramammary tubes for lactating cows, three defined daily doses were assumed to be 1 DCD (nDCD_lactation_), again regardless of antimicrobial active ingredient or concentration [33] (3). The active substance pirlimycine was not used on any of the dairy herds during the study period.
Total amount of active substance (mg)  nDCD_injectable_ =  (defined daily dose (mg/kg) per treatment course × concentration active substance (mg) × 425 kg)(1)
Total number of dry cow injectors,  nDCD_dry cow_ = 4(2)
Total number of intramammary tubes for lactating cows,  nDCD_lactation_ =  (defined daily dose (tubes per day) × 3)(3)
Total nDCD = nDCD_injectable_ + nDCD_dry cow_ + nDCD_lactation_(4)

#### 4.4.2. Somatic Cell Count and Milk Yield

Individual test-day SCC values (×1000 cells per mL) and composite MY values (kg of milk/d) per cow were available via DHI records, and were collected in a database from calving until 100 DIM (or culling).

#### 4.4.3. Clinical Mastitis and Culling

Producers were asked to record and sample the first CM case of each cow in the study, starting at drying off until 100 DIM (or until culling if the cow was culled before 100 DIM). Sample materials were provided and sampling techniques were verified to ensure aseptic sampling. Signs of CM were defined as visible abnormalities of the udder or milk (absence or presence of clots in milk, and a hard and painful quarter) or systemic illness, indicating udder inflammation. Additionally, all culling events between drying off and 100 DIM were recorded by the farmer. The reasons for culling were either udder health-related (mastitis or teat-end injuries) or related to other issues (lameness, abortion, old age, gastrointestinal problems, etc.).

### 4.5. Statistical Analyses

#### 4.5.1. Antimicrobial Consumption

Due to the data distribution, a non-parametric independent-samples Mann–Whitney U test was performed to identify differences with respect to the antimicrobial consumption expressed in DCD between the BDCT and the SDCT group. The analysis was performed using SPSS (version 26.0; SPSS Inc., Chicago, IL, US). Significance was assessed at *p* < 0.05. The data were presented in a histogram.

#### 4.5.2. Test-Day Somatic Cell Count and Milk Yield

The association between the treatment group (BDCT vs. SDCT, the predictor variable of main interest) and the natural logarithm of test-day SCC (LnSCC) and MY values (the outcome variables), respectively, was determined using linear mixed regression models (PROC MIXED, SAS 9.4, SAS Institute Inc., Cary, NC, USA). Herd and cow were included as random effects to correct for the clustering of cows in herds and repeated measurements within cows. The model also included DIM and the quadratic term for DIM as continuous predictor variables, and the treatment group (two levels: BDCT and SDCT) as the categorical predictor variable of main interest. LnSCC values for the outcome variable MY only (continuous variable), and parity (two levels: primiparous and multiparous cow) for both test-day SCC and MY values were initially included in the models as continuous and categorical predictor variables, respectively. Statistical significance was assessed at *p* < 0.05. The goodness of fit of the models was tested by examining the normal-probability plots of residuals and the plots of residuals vs. the predicted values to check whether or not the assumptions of normality and homogeneity of variance had been fulfilled. No patterns indicating heteroscedasticity were revealed.

#### 4.5.3. Clinical Mastitis and Culling

The association between the treatment groups (SDCT vs. BDCT, the predictor variable of main interest), the risk of developing CM and the hazard of being culled (outcome variables), respectively, was determined using survival analysis. A shared frailty survival model was fit with the time in days between drying off and the occurrence of CM (cows with the first case of CM vs. censored cows (at 100 DIM or until culling)) or a culling event (culled vs. censored cows (at 100 DIM)) as outcome variables. In the case of no occurrence of CM or culling, cows were censored at 100 DIM. Treatment group (two levels: BDCT and SDCT) and parity (two levels: primiparous and multiparous) were included as categorical predictor variables. Herd was included as a frailty effect to correct for the clustering of cows within herds (PROC PHREG, SAS 9.4, SAS Institute Inc.). Statistical significance was assessed at *p* < 0.05.

## 5. Conclusions

The implementation of SDCT on commercial dairy herds, following an algorithm that takes into account the herd’s BMSCC, DHI information and the CM history of each cow, had no substantial negative impact on udder health, MY and culling hazard while substantially reducing antimicrobial usage. Therefore, without making use of bacteriological culturing, SDCT can be successfully implemented in commercial dairy herds.

## Figures and Tables

**Figure 1 antibiotics-12-00901-f001:**
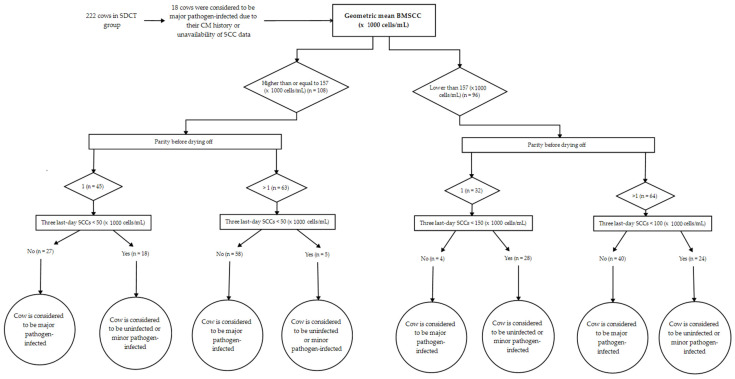
The algorithm [8] was applied to estimate the infection status at drying off of 222 cows that were allocated to the selective dry cow therapy (SDCT) group, taking into account the estimated herd level prevalence of subclinical mastitis (i.e., geometric mean bulk milk somatic cell count— (BMSCC)) and the cow’s parity. The herd’s geometric mean BMSCC was calculated over a 6-month period before drying off the cow (Milk Control Center Flanders, Lier, Belgium), while the DHI recordings provided SCC data and the parity of each cow (CRV, Sint-Denijs-Westrem, Belgium).

**Figure 2 antibiotics-12-00901-f002:**
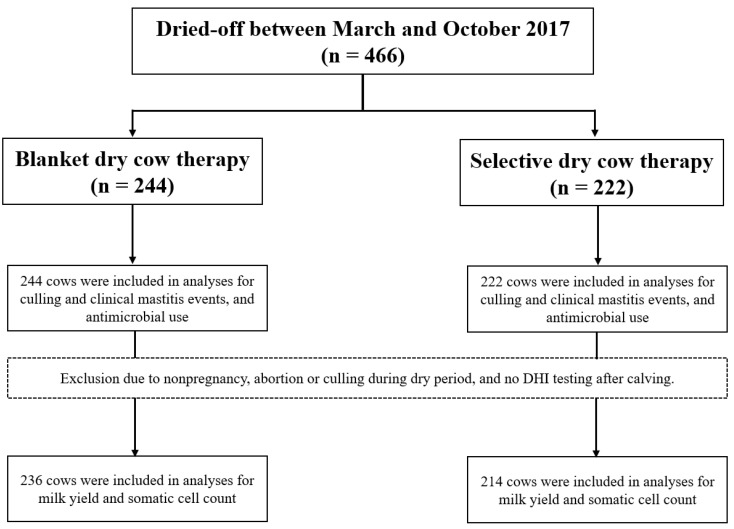
Cows that entered the field study received blanket dry cow therapy or selective dry cow therapy, according to their ear tag number (odd or even, respectively). Several analyses were performed, after excluding cows from which the required data was not available.

**Figure 3 antibiotics-12-00901-f003:**
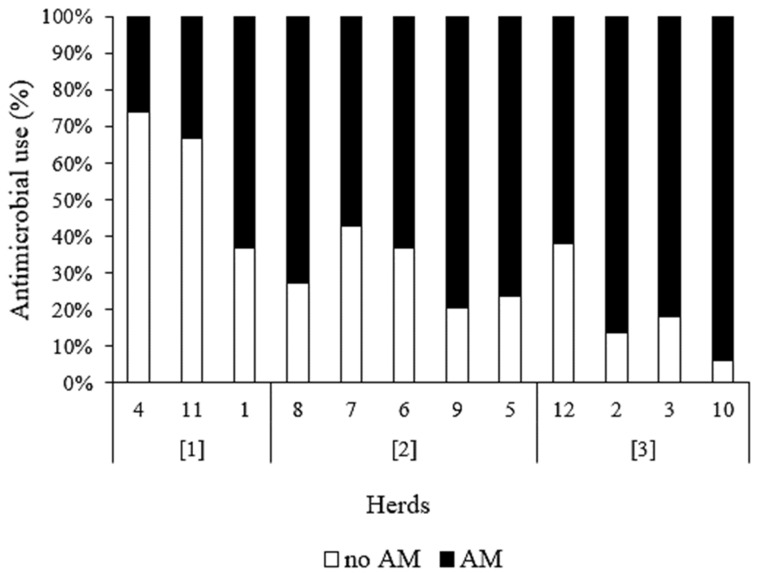
The proportion of cows in each herd that received antimicrobials (AM) or not (no AM) at drying off, according to their infection status as predicted by algorithm of Lipkens et al. [8], which is depicted in Figure 2. The 12 herds were sorted based on their geometric mean bulk milk SCC over a six-month period, which was recalculated monthly during the study from March until October 2018, and was used to stratify the herds into (1) herds with a low BMSCC (calculated geometric mean BMSCC over 6 months < 157,000 cells/mL), (2) herds that shifted from a low to a high BMSCC (calculated geometric mean BMSCC over 6 months ≥ 157,000 cells/mL) within the herd, and (3) herds with a high BMSCC.

**Figure 4 antibiotics-12-00901-f004:**
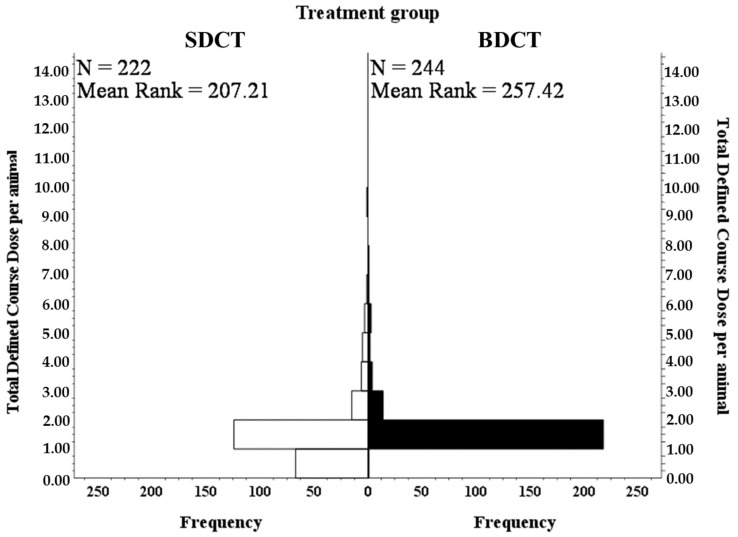
Representation of the frequency of the total defined course dose and the mean rank difference for selective dry cow therapy (SDCT) vs. blanket dry cow therapy (BDCT) using the non-parametric independent samples Mann–Whitney U test.

**Figure 5 antibiotics-12-00901-f005:**
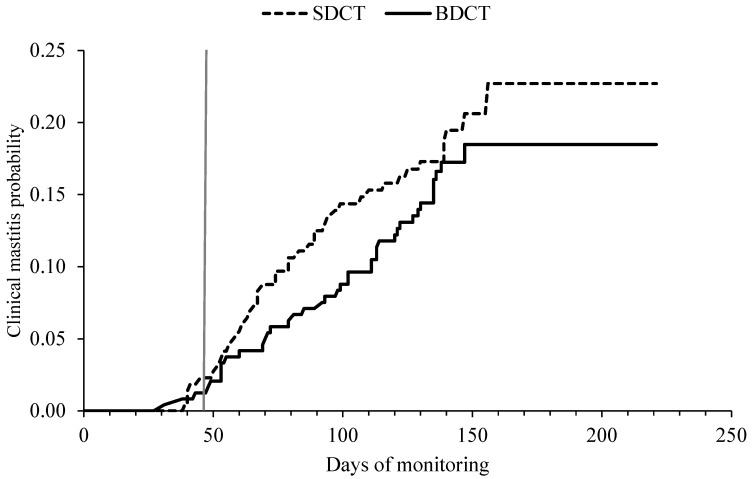
Kaplan–Meier graph showing the association between the treatment groups (selective dry cow therapy (SDCT) vs. blanked dry cow therapy (BDCT), always in conjunction with internal teat sealants) and the clinical mastitis hazard from the time of drying off until 100 days in milk in the subsequent lactation. The vertical grey line represents the average dry period length (46 days) of the 466 cows in the study.

**Figure 6 antibiotics-12-00901-f006:**
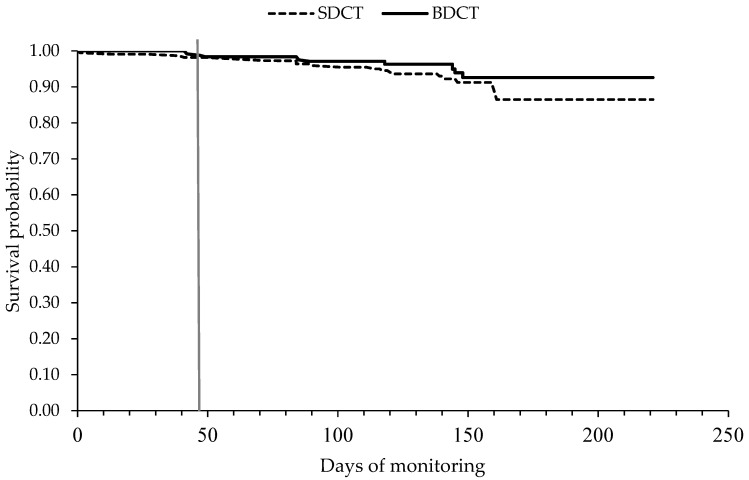
Kaplan–Meier graph showing the association between the treatment group (selective dry cow therapy (SDCT) vs. blanked dry cow therapy (BDCT), always in conjunction with internal teat sealants) and the clinical mastitis hazard from time of drying off until 100 DIM in the subsequent lactation. The vertical grey line represents the average dry period length (46 days) of the 466 cows in the study.

**Table 1 antibiotics-12-00901-t001:** Overview of herd characteristics and performance per herd of twelve commercial dairy herds in the Flemish region of Belgium.

Herd	Milking Parlor	Antimicrobial Dry Cow Therapy		Cows, *n*			Mean Dry Period Length (d) ^4^	Mean Milk Yield at the Last Test-Day Before Drying off (kg)	Geometric Mean BMSCC (×1000 cells/mL) ^5^	Mean 305 d Milk Production (kg)
		Antimicrobial Compound	Spectrum	Herd Size ^1^	Total in Field Study ^2^	Total (%) in SDCT Group ^3^				
Herd 1	Side-by-side	Cefquinome	Broad	110	48	19 (39.6)	37	22.4	128	9070
Herd 2	Herringbone	Cloxacillin and ampicillin	Broad	92	41	22 (53.7)	41	25.9	195	9571
Herd 3	Tandem	Cefazolin	Narrow	65	20	11 (55.0)	60	20.8	173	8063
Herd 4	Herringbone	Cefazolin	Narrow	106	50	23 (46.0)	40	24.3	84	8311
Herd 5	Side-by-side	Penethamate, benethamine penicillin, and framycetin	Broad	93	41	17 (41.5)	42	23.6	155	10,092
Herd 6	Herringbone	Cloxacillin	Broad	70	36	19 (52.8)	53	20.8	128	8134
Herd 7	Herringbone	Cefquinome	Broad	58	32	14 (43.8)	48	25.7	121	7915
Herd 8	Herringbone	Cefquinome	Broad	56	30	11 (36.7)	46	31.1	139	12,567
Herd 9	Tandem	Cefquinome	Broad	139	69	34 (49.3)	45	29.5	140	10,435
Herd 10	Herringbone	Procaine benzylpenicillin, dihydrostreptomycin, and nafcillin	Broad	73	30	16 (53.3)	58	11.3	194	5514
Herd 11	Herringbone	Cloxacillin	Broad	77	33	15 (45.5)	60	22.9	108	11,097
Herd 12	Carousel	Cefazolin	Narrow	128	36	21 (58.3)	44	21.6	178	9856

^1^ Number of lactating cows that were eligible to participate in the study (*n* = 1067). ^2^ Number of cows that were dried off according to the study’s protocol (*n* = 466). ^3^ Number (and percentage) of cows in the field study in the selective dry cow group that were allocated to dry cow treatment according to the study’s protocol (*n* = 222). ^4^ Mean dry period length (in days) that cows were dried off according to the study’s protocol (*n* = 466). ^5^ Geometric mean bulk milk somatic cell count in last 6 months before enrollment in the study.

**Table 2 antibiotics-12-00901-t002:** Overview of cow characteristics prior to drying off per treatment group on twelve commercial dairy herds in the Flemish region of Belgium.

Item	Treatment Group		
	Selective Dry Cow Therapy	Blanket Dry Cow Therapy	*p*-Value
Geometric mean SCC before drying off (× 1000 cells/mL)			
Third-last DHI record	75	74	0.783 ^1^
Second-last DHI record	82	82	0.952 ^1^
Last DHI record	101	93	0.454 ^1^
Parity			0.250 ^2^
Mean	2.46	2.30	
Range	1–9	1–9	
Standard deviation	1.63	1.46	
CM between last DHI record and drying off	7 out of 222 cows	6 out of 244 cows	0.640 ^2^

^1^ *p*-values based on bivariate analysis (*t*-test) of the Ln of test-day SCC by treatment group. ^2^ *p*-values based on bivariate analysis (Fisher’s exact) of each variable by treatment group.

**Table 3 antibiotics-12-00901-t003:** Mean, total and percentage of antimicrobial use for udder health in DCD ^1^ from (and including) drying off and 100 DIM, for the blanket dry cow therapy (BDCT) and selective dry cow therapy (SDCT) group.

Treatment Group	Number of Cows	Antimicrobial Use Per Indication (DCD ^1^)			Total
		Drying Off	Mastitis Treatment during Dry Period	Mastitis Treatment during Lactation	
SDCT	222				
Mean		0.67	0.01	0.38	1.06
Median		1.00	0.00	0.00	1.00
Total		149	1.74	85	236
%		63.0	0.74	36.2	100
BDCT	244				
Mean		1.01	0.00	0.24	1.25
Median		1.00	0.00	0.00	1.00
Total		245	0.00	58	303
%		80.9	0.00	19.1	100

^1^ Defined course dose.

**Table 4 antibiotics-12-00901-t004:** Final mixed regression models describing the associations between the selective dry cow therapy (SDCT) and blanket dry cow therapy (BDCT) groups and test-day SCC (LnSCC^2^) during subsequent lactation until 100 days in milk (DIM).

Variables	Cows, *n*	Milk Recordings, *n*	LnSCC ^1^ Per mL				
			Β ^2^	SE	LSM ^3^	*p*-Value ^4^	95% CI ^5^
Fixed part							
Intercept	450	1150	4.02	0.18	-	< 0.001	3.65–4.39
Treatment group						0.89	-
BDCT	236	539	Referent	-	3.55	-	-
SDCT	214	611	−0.01	0.10	3.53	0.89	−0.21–0.18
DIM	450	1150	−0.029	0.005	-	<0.001	−0.04–−0.02
Quadratic DIM	450	1150	0.0003	0.00005	-	<0.001	0.0002–0.0004
Parity						<0.001	-
Primiparous cows	178	463	Referent	-	3.44	-	-
Multiparous cows	272	687	0.39	0.10	3.74	<0.001	0.19–0.59

^1^ Natural log-transformed test-day SCC per mL. ^2^ Regression coefficient. ^3^ Least square means. ^4^ Overall *p*-value of the fixed effect. ^5^ 95% confidence interval.

**Table 5 antibiotics-12-00901-t005:** Final mixed regression models describing the associations between the selective dry cow therapy (SDCT) and blanket dry cow therapy (BDCT) groups and milk yield during subsequent lactation until 100 days in milk (DIM).

Variables	Cows, *n*	Milk Recordings, *n*	Milk Yield (kg/day)				
			Β ^1^	SE	LSM ^2^	*p*-Value ^3^	95% CI ^4^
Fixed part							
Intercept	450	1150	36.15	2.25	-	<0.001	31.39–40.92
Treatment group						0.54	
BDCT	236	539	Referent	-	38.40	-	-
SDCT	214	611	0.34	0.56	38.75	0.54	−0.76–1.45
DIM	450	1150	0.25	0.02	-	<0.001	0.20–0.30
Quadratic DIM	450	1150	−0.002	0.0002	-	<0.001	−0.003–−0.002
Parity						<0.001	-
Primiparous cows	178	463	Referent	-	36.84	-	-
Multiparous cows	272	687	3.47	0.58	40.31	<0.001	2.32–4.61
LnSCC ^5^	450	1150	−1.25	0.15	-	<0.001	−1.54–−0.95

^1^ Regression coefficient. ^2^ Least square means. ^3^ Overall *p*-value of the fixed effect. ^4^ 95% confidence interval. ^5^ Natural logarithm of test-day SCC.

**Table 6 antibiotics-12-00901-t006:** Final survival models describing the association between the selective dry cow therapy (SDCT) and blanket dry cow therapy (BDCT) groups and the clinical mastitis and culling hazard after drying off until 100 days in milk.

Outcome Variable	Predictor Variables	Cows (*n*)	Event (*n*, %)	β	SE	HR ^1^	95% CI HR ^2^	*p*-Value ^3^
Clinical mastitis ^4^								
	Treatment group							0.48
	BDCT	244	40 (16.4)	Referent				
	SDCT	222	42 (18.9)	0.16	0.22	1.17	0.76–1.81	
	Parity							0.004
	Primiparous	178	19 (11.6%)	Referent				
	Multiparous	288	63 (21.9%)	0.77	0.27	2.16	1.28–3.63	
Culling ^5^								
	Treatment group							0.24
	BDCT	244	13 (5.3)	Referent				
	SDCT	222	19 (8.6)	0.42		1.52	0.75–3.08	
	Parity							
	Primiparous	178	4 (2.2%)	Referent				0.01
	Multiparous	288	28 (11.5%)	1.40	0.54	4.05	1.42–11.63	

^1^ Hazard ratio. ^2^ 95% confidence interval of hazard ratio. ^3^ Overall *p*-value of the fixed effect. ^4^ A shared frailty term for the variable herd was included in the model (*p* < 0.001). ^5^ A shared frailty term for the variable herd was included in the model (*p* < 0.001).

**Table 7 antibiotics-12-00901-t007:** Overview of the number of clinical and culling events from dry off until 100 days in milk, the total defined course doses for the treatment of mastitis in the first 100 days in milk in the next lactation (nDCD_lactation_) ^3^, the average somatic cell count, and the average daily milk yield in the first 100 DIM stratified for the treatment group (blanket dry cow therapy (BDCT) ^1^ vs. selective dry cow therapy (SDCT) ^2^) and for antimicrobial consumption at dry off.

Parameter	BDCT ^1^		SDCT ^2^		
	Total (%)	AB ^3^ (%)	Total (%)	AB (%)	No AB ^4^ (%)
Culling	13/244 (5.3)	13/13 (100)	19/222 (8.6)	18/19 (94.7)	1/19 (5.3)
Clinical mastitis	40/244 (16.4)	40/40 (100)	42/222 (18.9)	31/42 (73.8)	11/42 (26.2)
nDCD_lactation_ ^5^	58	58	85	55	30
Average somatic cell count (×1000 cells/mL)	148	148	172	211	99
Average milk yield (kg/day)	38.8	38.8	39.1	39.2	39.1

^1^ Blanket dry cow treatment group. All cows received antimicrobials at dry off in this group. ^2^ Selective dry cow treatment group. ^3^ Cows within either BDCT or SDCT group that received antimicrobials. ^4^ Cows within SDCT group that received no antimicrobials. ^5^ Total number of defined course doses used for the treatment of mastitis during the first 100 days in milk during the next lactation.

## Data Availability

The data presented in this study are available upon request from the corresponding author. The data are not publicly available due to privacy reasons. The details of the dairy producers are listed in the database, and even through the unique identification number of each animal, the details of each dairy producer can be traced.

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
