# Peer review of "Impact of Selective Dry Cow Therapy on Antimicrobial Consumption, Udder Health, Milk Yield, and Culling Hazard in Commercial Dairy Herds"

_antibiotics, 2023, doi:10.3390/antibiotics12050901_

Round 1

Reviewer 1 Report

As the authors themselves noted, the number of animals, housing conditions, type of barn, nutrition, productivity and many others have a significant impact on the results.

My comments are as follows:

1. Why does the results section come before material and methods?

2. Some of the information from the materials and methods chapter can be found in the results... Like, for example, the type of milking machine... I suggest re-editing it. After deleting and organizing these 2 sections, the tables will become more readable.

3. 12 different herds - can they be compared in terms of health? Vaccinations, BCS, welfare, ect…? There is no such information. This has a significant impact on the immune system.

4. The research material was collected from March 2017 till April 2018. Why was it written so late? It's been ~ 5 years?

Author Response

  1. Why does the results section come before material and methods?

AU: We followed the template and instructions proposed by Antibiotics. According to these instructions, the Materials & Methods section is at the end of the manuscript.

2. Some of the information from the materials and methods chapter can be found in the results... Like, for example, the type of milking machine... I suggest re-editing it. After deleting and organizing these 2 sections, the tables will become more readable.

AU: Thank you very much for this suggestion. Indeed, some information could be moved from the Result section to the Materials and Methods section. The Herd Descriptives Statistics were also re-edited. The changes are highlighted via Track Changes in the revised manuscript (see L95-103; L653-654; L674-676). Still, as mentioned above, the Materials and Methods section is still at the end of the manuscript according to the Antibiotics instructions and recommendations.

3. 12 different herds - can they be compared in terms of health? Vaccinations, BCS, welfare, ect…? There is no such information. This has a significant impact on the immune system.

AU: Thanks for this comment. In some way, the farms can be compared in terms of health. As mentioned in the Materials and Methods section, all herds had a geometric mean BMSCC ≤ 250,000 cells/mL for a period of six months before enrollment in the study. This inclusion criterium selects already for the better farms when it comes to udder health, and make them more comparable in terms of animal health and management than the average farm in the Flemish region of Belgium. Also, all 12 herds were approved by the Belgian Dairy Quality Assurance Program (IKM) and thus comply with the legal standards regarding animal welfare, environment & hygiene, the purity & safety of the final milk product, the avoidance of the introduction of foreign substances (for example antimicrobial residues) in the milk. Nevertheless, we do not have such detailed information regarding vaccination programs and BCS. Those factors might indeed have an impact on the immunity of the cow and thus the risk of new infections and the cure rates over dry period. Still, we like to emphasize that per farm the cows assigned to the blanket dry cow group and the cows assigned to the selective dry cow group were housed under the same management conditions. It is not that blanket dry cow therapy was applied on one farm and selective dry cow therapy on another farm. On all farms, both dry-off strategies were applied. This study design was selected in order to exclude variation among farms as much as possible and avoid confounding factors linked to the cow's immunity as mentioned by the reviewer. Also, in all statistical models, farm was included as random effect in order to take into account clustering of cows within a farm. 

4. The research material was collected from March 2017 till April 2018. Why was it written so late? It's been ~ 5 years?

AU: Indeed, the research was part of a PhD study. This study was actually the last part of the PhD research project of the first author. As this is often the case, not all manuscripts of all studies could be finished by the end of the PhD project. The data analyses and manuscript writing were finalized by the PhD candidate while she was already working somewhere else. This delayed the writing process and thus also the submission of this publication. 

Reviewer 2 Report

Udder health is the key to an efficient activity. Its permanent and exact supervision is desirable in dairy farms. The work done by you is hard and must be rewarded accordingly. Although there are numerous studies in the field, this study should not be ignored.

Author Response

Thank you very much for the reviewer's positive feedback. 

Reviewer 3 Report

Comments to the Authors

At the outset, I would like to thank you for the opportunity to participate in the review of the manuscript entitled “Impact of selective dry cow therapy on antimicrobial consumption, udder health, milk yield, and culling hazard in commercial dairy herds” evaluated whether implementing selective dry cow therapy (SDCT) on commercial dairy farms reduces antimicrobial consumption without negatively affecting future performances when compared to blanket dry cow therapy (BDCT). 

The topic of this paper was relevant, timely, and of interest to the audience of this journal.

The content of this paper was technically accurate and sound.

I consider the entire manuscript interesting and worthy of attention. The manuscript was pleasant to read.

Nevertheless, I believe that this work deserves publication after the inclusion of some minor editing of English language.

I believe that this work deserves publication after the inclusion of some minor editing of English language.

Author Response

AU: Thank you very much for your positive feedback on our manuscript. The manuscript has been checked by a native English speaking colleague.